# SKIP TO THE GOOD PART: REPRESENTATION STRUCTURE & INFERENCE-TIME LAYER SKIPPING IN DIFFUSION VS. AUTOREGRESSIVE LLM

**Raghavv Goel**[*], **Risheek Garrepalli,**[*]**Sudhanshu Agrawal, Chris Lott, Mingu Lee**[γ]**, Fatih Porikli**[γ]
Qualcomm AI Research
San Diego, USA
{raghgoel,rgarrepa,mingul}@qti.qualcomm.com

## ABSTRACT

Autoregressive (AR) language models form representations incrementally through left-to-right prediction, whereas diffusion language models (dLLMs) are trained via full-sequence denoising. We perform the first layer- and token-wise representational analysis comparing native dLLMs (LLaDA), native AR models (Qwen2.5), and AR-initialized dLLMs (Dream-7B). We find that diffusion objectives produce hierarchical abstraction with early-layer redundancy and reduced recency bias, while AR objectives yield tightly coupled, depth-dependent representations. Critically, AR-initialized dLLMs retain AR-like dynamics despite diffusion training, revealing persistent initialization bias. Leveraging this redundancy, we introduce static, task-agnostic inference-time layer-skipping requiring no architectural changes. Native dLLMs achieve up to 18.75% FLOPs reduction while preserving over 90% performance on reasoning and code generation benchmarks, whereas AR models degrade sharply under comparable skipping.

## 1 INTRODUCTION

Autoregressive (AR) LLMs trained with next-token prediction build predictions left-to-right, while diffusion LLMs (dLLMs) learn by iteratively denoising full sequences. Although recent dLLMs (e.g., LLaDA Nie et al. (2025), DiffuCoder Gong et al. (2025)) match AR performance, how their internal representations differ remains unexplored. While dLLM research focuses on inference efficiency—parallel decoding, KV-caching, architectural optimizations—the representational properties remain largely unexamined. Recent work Gong et al. (2025) has begun examining local versus global representations in dLLMs, but *systematic analysis of how diffusion objectives shape representational abstraction across layers compared to AR models is still missing*.

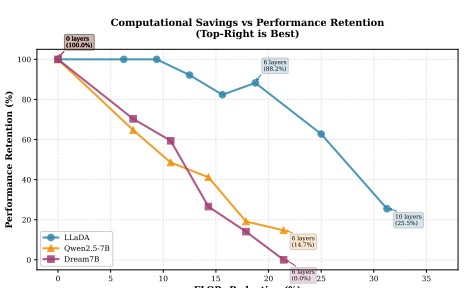

Figure 1: **Representational redundancy enables efficient inference-time layer skipping.** We evaluate zero-shot layer pruning across three architectures: LLaDA (diffusion LLM), Dream7B (dLLM initialized from Qwen2.5), and Qwen2.5-7B (AR-LLM). The diffusion-based LLaDA exhibits remarkable robustness, retaining 88.24% performance at 18.75% FLOPs reduction (6 layers skipped), demonstrating significant representational redundancy. Conversely, autoregressive models show brittle behavior with only 64.71% retention at 7.14% FLOPs reduction (2 layers), revealing concentrated, non-redundant representations. Top-right region indicates optimal performance-efficiency trade-off.

We hypothesize that training objectives materially shape internal geometry, leading diffusion-trained models to organize information differently. To isolate the role of objective and initialization, we compare three families: native dLLM (LLaDA), native AR (Qwen2.5), and AR-adopted dLLM (Dream-7B). Our analysis shows Dream-7B aligns more with Qwen2.5 than LLaDA, indicating persistent AR initialization structure. We assess whether

---

[*]Qualcomm AI Research is an initiative of Qualcomm Technologies, Inc

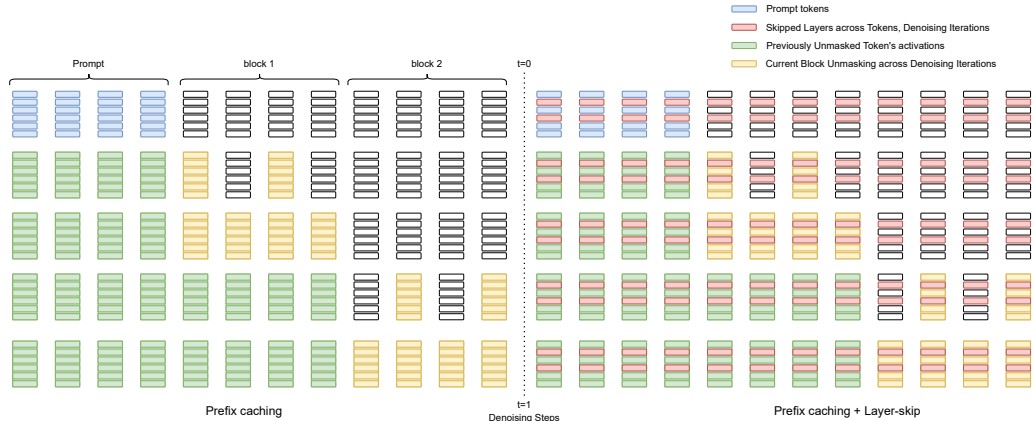

Figure 2: **Layer-skip mechanism for dLLMs.** At each denoising step, high-similarity layers (shaded) are bypassed, with hidden states passed directly to the next active layer. This reduces per-step FLOPs while preserving the coarse-to-fine abstraction hierarchy.

observed redundancy enables static, task-agnostic layer skipping at inference time—without KV sharing—to yield speedups with minimal performance loss. Our contributions:

- *Representational analysis:* First systematic layer-wise and token-wise comparison showing diffusion objectives produce hierarchical abstraction with early-layer redundancy and minimal recency bias, while AR objectives yield incremental refinement and strong recency bias. AR-initialized dLLMs retain AR-like structure despite diffusion training.

- *Inference-time layer skipping:* Static, task-agnostic layer-skip policy requiring no architectural changes. Native dLLMs achieve up to 18.75% FLOPs reduction with ¡10% accuracy degradation; AR models show brittleness.

- *Cross-domain benchmarking:* Native dLLMs tolerate aggressive skipping (6+ layers with ¡90% retention), AR-initialized dLLMs show moderate robustness (2–4 layers), AR models degrade substantially.

## 2 LAYER-WISE AND TOKEN-WISE SIMILARITY ANALYSIS

To understand how training objectives shape internal representations, we examine layer-to-layer similarity across dLLMs and AR models. We track cosine similarity between consecutive layer representations $\mathbf{h}_\ell$ and $\mathbf{h}_{\ell+1}$ across tokens. For dLLMs, we compute this at multiple denoising steps $t \in \{1, \ldots, T\}$; for AR models, during standard forward passes.

**Key Findings: (1) Coarse-to-fine abstraction in native dLLMs:** LLaDA (Fig. 5) shows consistent layer-wise similarity across denoising steps. Early layers establish coarse representations with high inter-layer similarity ($> 0.95$), while later layers perform iterative refinement, indicating exploitable early-layer redundancy.

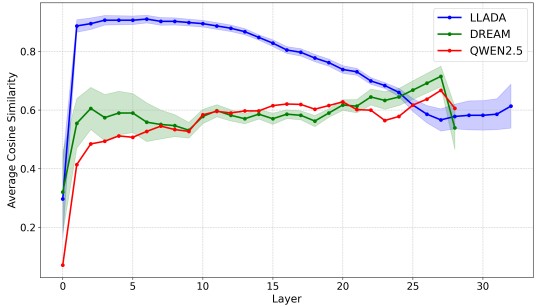

Figure 3: **Average token-wise cosine similarity across layers and denoising steps.** LLaDA (native dLLM) exhibits high similarity ($> 0.9$) in early layers with smooth transitions, followed by lower similarity in later layers where refinement occurs. Dream-7B closely follows Qwen2.5's pattern despite diffusion training, revealing persistent initialization bias. Shaded regions show standard deviation across denoising steps for LLaDA and Dream-7B.

**(2) Recency bias differences:**
LLaDA exhibits minimal recency bias with smooth, high-similarity transitions across tokens (global representations) as shown in Figure 4. Dream-7B and Qwen2.5 show significant recency bias—substantial representational changes for each new token across all layers. In LLaDA, recency bias emerges only in later layers; in AR models, it persists throughout depth, indicating less hierarchical abstraction.

**(3) Strong initialization bias:** Despite diffusion training, Dream-7B's similarity profile mirrors Qwen2.5's AR initialization. Figure 3 shows Dream-7B's token-wise cosine similarity follows Qwen2.5 throughout network depth. LLaDA exhibits distinctly different patterns: very high similarity ($> 0.95$) in early layers (redundant representations), transitioning to lower similarity in later layers (active refinement)—a hierarchical pattern absent in AR-initialized models.

---

### Key Takeaways

**Diffusion objectives reduce recency bias and promote hierarchical abstraction:** LLaDA shows minimal recency bias with global representations; AR models exhibit strong recency bias at all layers.

**AR initialization creates persistent structure:** Dream-7B retains Qwen2.5's similarity patterns despite diffusion training, demonstrating strong regularization from pre-trained AR representations.

---

## 3 LAYER-SKIP AT INFERENCE

High-similarity plateaus suggest certain layers contribute minimally to representational transformation.

We hypothesize *skipping* these layers reduces computational cost with minimal performance impact (Fig. 2). Our approach is: (1) *Static and task-agnostic:* Skip-eligible layers identified from training-time similarity without per-task tuning. (2) *Architecture-agnostic:* Applies to any pretrained model without modification. (3) *Complementary to KV-caching:* Layer skipping reduces FLOPs/depth; KV-caching reduces memory.

**Skip policy:** Based on Figure 5, we define skip set $\mathcal{S} \subset \{1, \ldots, L\}$ containing layers with consecutive similarity $> \theta$ ($\theta = 0.95$). During inference, for $\ell \in \mathcal{S}$, we bypass the transformer block and pass $\mathbf{h}_{\ell-1}$ directly to layer $\ell + 1$. Algorithm in Appendix Section 8; magnitude analysis in Appendix Section 10.1.

**Hypothesis:** Minimal degradation for dLLMs (high similarity indicates redundancy); larger degradation for AR models (rely on incremental refinement).

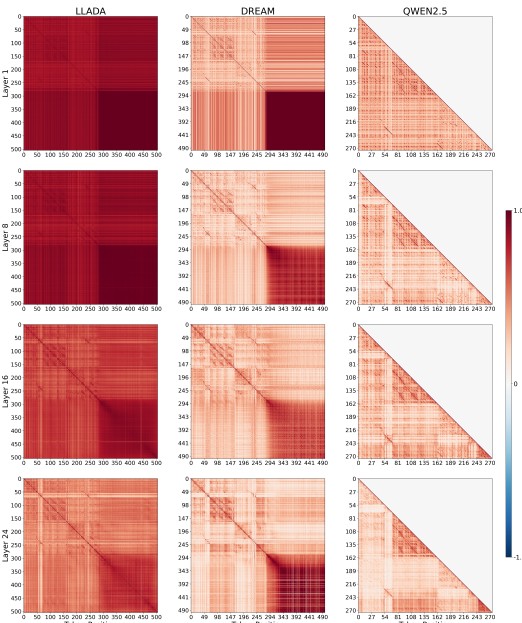

Figure 4: **Token-wise cosine similarity across layers and models.** Rows: layers $1, 8, 16, 24$; columns: LLaDA (left), Dream-7B (middle), Qwen (right). Limiting decoding to 32 tokens highlights early stabilization in native diffusion models.

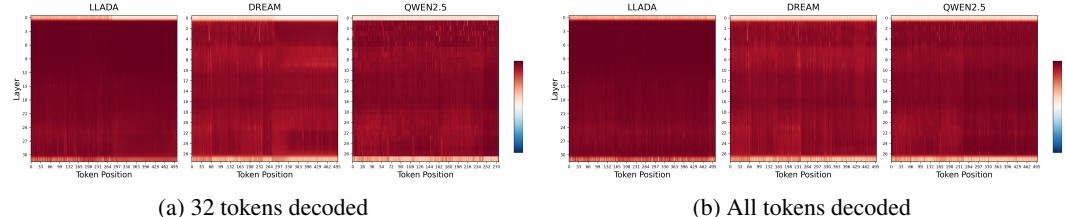

(a) 32 tokens decoded                (b) All tokens decoded

Figure 5: **Layer-wise cosine similarity across models.** Each row shows similarity between consecutive layers for (top) LLaDA, (middle) Qwen2.5, and (bottom) Dream-7B. High-similarity regions (yellow) indicate representational redundancy. Dream-7B's pattern closely resembles Qwen2.5 despite diffusion training, revealing strong initialization bias.

## 4 RESULTS

We evaluate three model families (LLaDA, Dream-7B, Qwen2.5) across four benchmarks (GSM8K, MATH-500, HumanEval, MBPP). Detailed experimental setup is provided in Appendix Section 7.

*Native dLLMs Enable Aggressive Layer Skipping.* Table 1 and Figure 1 show that native diffusion LLMs are markedly more robust to layer skipping than autoregressive models. For LLaDA, skipping 6 layers (18.75% FLOPs reduction) preserves 88.2–102.1% of baseline performance across all evaluated tasks. Even at an 8-layer skip (25% FLOPs reduction), performance retention remains high (62.7–91.8%), placing LLaDA firmly in the favorable efficiency–quality regime (top right of Figure 1).

In contrast, autoregressive models degrade rapidly under similar interventions. Qwen2.5 experiences severe performance collapse: skipping only 2 layers (7.14% FLOPs reduction) reduces retention to 34.9–64.7%, indicating that AR representations are tightly coupled and lack depth-wise redundancy. Notably, Dream-7B—despite diffusion fine-tuning—exhibits similar brittleness. At a 2-layer skip, Dream-7B achieves only 60.5–81.4% retention, closely tracking Qwen2.5 and substantially underperforming LLaDA (100–123.1%). This provides strong evidence that autoregressive pre-training induces a persistent representational structure that diffusion objectives alone do not override, preventing the emergence of the coarse-to-fine hierarchy observed in native dLLMs. Full table with MATH500 and MBPP is present in Appendix Section 9 Table 2.

**Computational savings:** Native dLLMs achieve **2.6× greater FLOPs reduction with 1.4× higher quality retention**. Importantly, these savings are orthogonal to KV-caching: layer skipping reduces depth-wise computation, while KV-caching eliminates token-wise redundancy, enabling multiplicative gains when combined.

Table 1: Performance comparison across models and layer skipping configurations. Values represent retention percentages relative to baseline (0 layers skipped), with absolute accuracy shown in parentheses for baseline rows. Missing entries (−) indicate performance below retention threshold. Higher values (↑) indicate better retention.

| Benchmark | Layers Skipped | Llada-Instruct | Dream-Base | Dream-Instruct | Qwen2.5 |
|---|---|---|---|---|---|
| **GSM8K** | 0 (baseline) | 100.0 (0.83) | 100.0 (0.77) | 100.0 (0.81) | 100.0 (0.49) |
| | 2 | 101.3 | 76.8 | 84.6 | 34.9 |
| | 3 | 96.7 | 62.6 | 71.2 | 15.9 |
| | 4 | 102.5 | 44.5 | 25.0 | 20.6 |
| | 5 | 97.7 | − | − | − |
| | 6 | 91.8 | − | − | − |
| | 8 | 91.8 | − | − | − |
| **HumanEval** | 0 (baseline) | 100.0 (0.51) | 100.0 (0.65) | 100.0 (0.64) | 100.0 (0.68) |
| | 2 | 100.0 | 66.2 | 70.3 | 64.7 |
| | 3 | 100.0 | 63.1 | 59.4 | 48.5 |
| | 4 | 92.2 | 43.1 | 26.6 | 41.2 |
| | 5 | 82.3 | − | − | − |
| | 6 | 88.2 | − | − | − |
| | 8 | 62.7 | − | − | − |

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

## 5 APPENDIX

## 6 RELATED WORK

**Diffusion Language Models and Representations.** Diffusion language models (dLLMs) replace autoregressive decoding with bidirectional denoising objectives, enabling parallel decoding and global context modeling. Foundational work on discrete diffusion Austin et al. (2021) led to recent dLLMs such as SEDD Lou et al. (2023), and LLaDA Nie et al. (2025); Bie et al. (2025), which achieve competitive language modeling performance. Dream-7B Ye et al. (2025) adapts pretrained AR models to diffusion training, while MDLM Sahoo et al. (2024) simplifies diffusion objectives. Despite this progress, how diffusion objectives shape internal representations—especially relative to AR and AR-initialized models—remains insufficiently understood. Additionally, efforts have been made to integrate KV caching mechanisms to reduce redundant computation Ma et al. (2025); Liu et al. (2025). An alternate line of work focuses on step distillation of dLLMs Deschenaux & Gulcehre (2025); Qian et al. (2026) towards accelerating dLLM inference.

**Representational Structure and Initialization Effects.** Prior work shows that language models organize representations hierarchically across depth, with earlier layers capturing coarse features and deeper layers refining task-specific abstractions Jawahar et al. (2019). While representational dynamics in AR models have been studied, systematic analyses comparing AR and diffusion models are rare. Our work provides direct, layer- and token-level evidence of this initialization bias in AR-adapted dLLMs and contrasts it with the representational redundancy that emerges in native diffusion models.

## 7 EXPERIMENTAL SETUP

**Models.** We evaluate three families to disentangle training objective and initialization: (i) a native diffusion LLM, **LLaDA** (we use the 8B Base / Instruct checkpoints) (Nie et al., 2025); (ii) a native autoregressive (AR) model, **Qwen2.5** (7B Base / Instruct) (Yang et al., 2024); and (iii) an AR-initialized diffusion LLM, **Dream-7B** (Instruct) (Ye et al., 2025). Unless stated otherwise, all models are evaluated with their public inference code and default tokenizers.

**Benchmarks.** We measure reasoning and code synthesis across standard suites: **GSM8K** (grade-school math; exact-match accuracy) (Cobbe et al., 2021); **HumanEval** (function-level Python synthesis; pass@k using the official harness) (Chen et al., 2021); **MATH-500** (the 500-problem test subset of the MATH benchmark; exact-match accuracy) (Hendrycks et al., 2021).

**Prompting and answer extraction.** For **GSM8K**, we use a few-shot rationale prompt with an explicit `Final Answer:` line; we strip formatting and compare normalized numbers. For **HumanEval** and **MBPP**, we request a single Python function and evaluate with the official test suites; we report pass@1 and pass@k. The same prompts are used across models to ensure comparability. We follow exact inference setting of Chen & Liu (2026)

**Decoding & sampling.** For **AR** decoding (Qwen2.5), we use greedy or nucleus sampling (default `top_p`=0.95, `temperature`$\in \{0.2, 0.7, 0.8\}$ depending on task), `max_new_tokens`=2048, and early stopping on task-specific end markers. For **diffusion** decoding (LLaDA, Dream-7B), we follow each repository's default sampler/schedule and report quality–latency tradeoffs across denoising budgets $T \in \{16, 32\}$; other settings (e.g., temperature annealing or remasking) follow the public implementations (Nie et al., 2025; Ye et al., 2025). To compare fairly to AR decoding, we standardize context limits (2,048 tokens total) and stop rules.

**Layer-skipping evaluation (ours).** To isolate objective-induced redundancy, we introduce a *static, task-agnostic* top-$k$ layer skip policy applied only at inference time, without KV sharing or architectural changes. We evaluate $k \in \{0, 2, 4, 6\}$ on 7–8B models. Metrics include: task score (GSM8K accuracy; HumanEval/MBPP pass@1 and pass@k; MATH-500), *equivalence rate* (fraction of generations identical to full-depth outputs), token-level KL divergence (w.r.t. the full-depth distribution; optional), and end-to-end latency/throughput (prefill+decode or diffusion steps).

*Note:* Due to varying inference optimizations across model implementations ( different kernel implementations), absolute throughput (tokens/sec) is not directly comparable between models. We therefore report: (1) the number of layers skipped for each model, and (2) the resulting task performance relative to full-depth baselines. This isolates the effect of our layer-skip strategy from architecture-specific optimizations.

## 8  LAYER-SKIP ALGORITHM

---
**Algorithm 1** Layer Skipping Algorithm

---
1: $\mathcal{S} \leftarrow [], \mathcal{L}_{\text{skip}} \leftarrow \emptyset, \tau \leftarrow \text{threshold}, n_{\max} \leftarrow \text{max layers to skip}$    $\triangleright \mathcal{S}$: similarity list, $\mathbf{H}$: hidden states, $p$: prompt length
2: **for** $i \leftarrow 1$ to $|\mathbf{H}| - 1$ **do**
3:      $\mathbf{h}_{\text{prev}} \leftarrow \mathbf{H}_{i-1}[:, : p]$
4:      $\mathbf{h}_{\text{curr}} \leftarrow \mathbf{H}_i[:, : p]$
5:      $s \leftarrow \text{cosine\_similarity}(\mathbf{h}_{\text{prev}}, \mathbf{h}_{\text{curr}}).\text{mean}()$
6:      $\mathcal{S}.\text{append}(s)$
7: **end for**                                       $\triangleright$ Sort by similarity (descending)
8:   $\_, \mathbf{idx} \leftarrow \text{sort}(\mathcal{S}, \text{descending} = \text{True})$ $\triangleright \mathbf{idx}$: sorted indices $\triangleright$ Apply rules to select which layers to skip
9: **for** $i \leftarrow 0$ to $|\mathbf{idx}| - 1$ **do**
10:      $l \leftarrow \mathbf{idx}[i]$                 $\triangleright$ Check if current layer is not adjacent to any already skipped layer
11:      **if** $l \notin \mathcal{L}_{\text{skip}}$ **and** $l - 1 \notin \mathcal{L}_{\text{skip}}$ **and** $l + 1 \notin \mathcal{L}_{\text{skip}}$ **and** $\mathcal{S}[l] \geq \tau$ **then**
12:          $\mathcal{L}_{\text{skip}}.\text{add}(l)$                 $\triangleright$ Limit the maximum number of layers to skip
13:          **if** $|\mathcal{L}_{\text{skip}}| \geq n_{\max}$ **then**
14:              **break**
15:          **end if**
16:      **end if**
17: **end for**                                    $\triangleright$ Prepare final list of layers to skip
18: $\mathcal{L}_{\text{skip}} \leftarrow \text{sorted}(\mathcal{L}_{\text{skip}})$        $\triangleright$ Add 1 to indices as we always skip the later layer in each pair
19: $\mathcal{L}_{\text{skip}} \leftarrow [l + 1 \text{ for } l \text{ in } \mathcal{L}_{\text{skip}}]$
20: **return** $\mathcal{L}_{\text{skip}}$

---

## 9  ADDITIONAL RESULTS

We provide full table with additional tasks: MATH500 and MBPP in Table 2.

### 9.1  LAYER DISTRIBUTION AND SKIP SENSITIVITY

Tables 3a and 3b reveal that *consecutive layer skipping is catastrophic*. For LLaDA at 6-layer skip, allowing consecutive removal drops GSM8K retention from 91.8% to 75.3% and HumanEval from 88.2% to 64.7%. Dream-7B shows extreme sensitivity: 2-layer consecutive skip collapses GSM8K retention from 76.8% to 14.1%.

Our algorithm avoids this by maintaining representational continuity. Analysis and Fig.6 shows skipped layers concentrate in early network depth (first 40–60%), aligning with our observation that early layers establish coarse representations with high redundancy, while later layers perform critical fine-grained refinement.

## 10  ADDITIONAL ANALYSIS AND VISUALIZATIONS

### 10.1  MAGNITUDE EVOLUTION

One potential limitation of cosine similarity is its invariance to magnitude. To ensure our redundancy findings are not artifacts of magnitude collapse, we analyze the $\ell_2$ norm of hidden states across layers as observed in Fig. 7.We observed that the magnitude evolution is small for initial 60-70% layers and

Table 2: Performance comparison across models and layer skipping configurations. Values represent retention percentages relative to baseline (0 layers skipped), with absolute accuracy shown in parentheses for baseline rows. Missing entries (−) indicate performance below retention threshold. Higher values (↑) indicate better retention.

| Benchmark | Layers Skipped | Llada-Instruct | Dream-Base | Dream-Instruct | Qwen2.5 |
|---|---|---|---|---|---|
| GSM8K | 0 (baseline) | 100.0 (0.83) | 100.0 (0.77) | 100.0 (0.81) | 100.0 (0.49) |
| | 2 | 101.3 | 76.8 | 84.6 | 34.9 |
| | 3 | 96.7 | 62.6 | 71.2 | 15.9 |
| | 4 | 102.5 | 44.5 | 25.0 | 20.6 |
| | 5 | 97.7 | − | − | − |
| | 6 | 91.8 | − | − | − |
| | 8 | 91.8 | − | − | − |
| MATH500 | 0 (baseline) | 100.0 (0.37) | 100.0 (0.34) | 100.0 (0.43) | 100.0 (0.18) |
| | 2 | 108.5 | 81.4 | 78.2 | − |
| | 3 | 89.4 | 60.5 | 54.6 | − |
| | 4 | 89.4 | 58.8 | 32.7 | − |
| | 5 | 93.6 | − | − | − |
| | 6 | 102.1 | − | − | − |
| | 8 | 70.2 | − | − | − |
| HumanEval | 0 (baseline) | 100.0 (0.51) | 100.0 (0.65) | 100.0 (0.64) | 100.0 (0.68) |
| | 2 | 100.0 | 66.2 | 70.3 | 64.7 |
| | 3 | 100.0 | 63.1 | 59.4 | 48.5 |
| | 4 | 92.2 | 43.1 | 26.6 | 41.2 |
| | 5 | 82.3 | − | − | − |
| | 6 | 88.2 | − | − | − |
| | 8 | 62.7 | − | − | − |
| MBPP | 0 (baseline) | 100.0 (0.52) | 100.0 (0.43) | 100.0 (0.71) | 100.0 (0.81) |
| | 2 | 123.1 | 60.5 | 70.3 | 75.3 |
| | 3 | 115.4 | 74.4 | 54.1 | 55.6 |
| | 4 | 115.4 | 53.5 | 37.8 | 35.8 |
| | 5 | 111.5 | − | − | − |
| | 6 | 94.2 | − | − | − |
| | 8 | 71.1 | − | − | − |

Table 3: Performance retention for models with layer-skip when **allowed** and **not allowed** to skip consecutive layers. Higher (↑) the better

| Layers Skipped | GSM8k | | HumanEval | |
|---|---|---|---|---|
| | Not Allowed | Allowed | Not Allowed | Allowed |
| 4 | 102.5 | 95.1 | 92.2 | 88.2 |
| 6 | 91.8 | 75.3 | 88.2 | 64.7 |
| 8 | 91.8 | 45.8 | 62.7 | 23.5 |

(a) **LLaDA** model

| Layers Skipped | GSM8k | | HumanEval | |
|---|---|---|---|---|
| | Not Allowed | Allowed | Not Allowed | Allowed |
| 2 | 76.8 | 14.1 | 78.2 | 67.7 |
| 3 | 62.6 | 18.2 | 54.6 | 53.8 |
| 4 | 44.5 | 16.2 | 32.7 | 23.1 |

(b) **Dream** model

then rises steeply. There is also presence of sink tokens – super high magnitude than the rest of the tokens, as discussed in Rulli et al. (2025).

Together, these analyses validate cosine similarity as a meaningful proxy which demonstrate potential redundancy in representations and motivate our inference-time layer-skipping strategy.

## 10.2 DETAILED TOKEN-WISE SIMILARITY ANALYSIS

We provide comprehensive token-wise similarity visualizations to complement the layer-wise analysis presented earlier. These reveal how hidden state representations evolve across tokens within individual layers, providing deeper insight into the recency bias and global vs. local representation patterns discussed in the main text.

Figure 10 shows token-wise cosine similarity across all 32 layers of LLaDA. Early layers (0–15) exhibit consistently high similarity (> 0.9) between consecutive tokens, indicating smooth representational transitions with minimal recency bias. This validates our hypothesis that native dLLMs establish stable global context in early layers. Later layers (16–31) show increased variability and

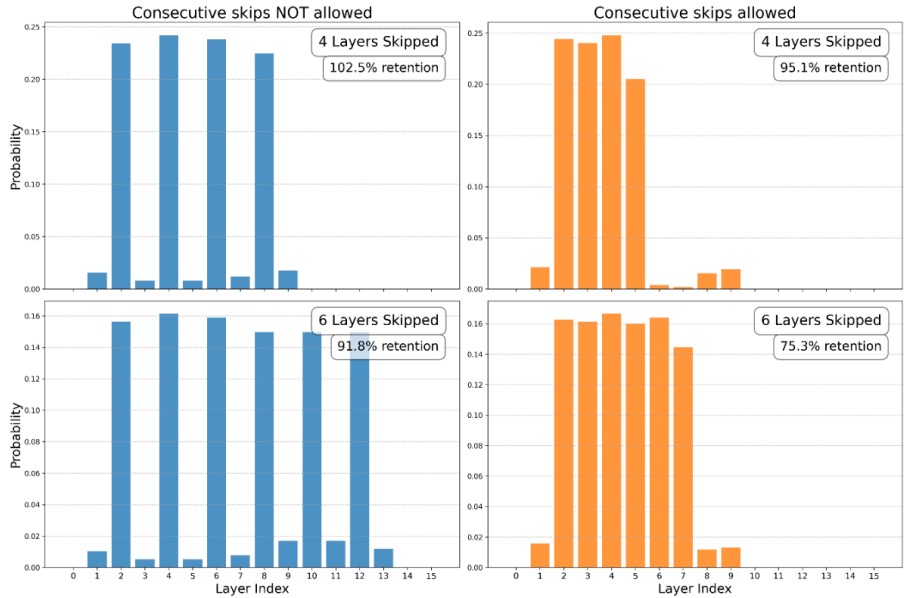

Figure 6: **Which layers are skipped?** Empirical distribution over *layer indices* selected for skipping by Algorithm 1on LLADA, aggregated across evaluation prompts on GSM8k.The plotted value at layer $i$ is the fraction (probability) of runs/examples in which layer $i$ is chosen to be skipped. **Left:** consecutive (adjacent) layers may be skipped, allowing contiguous skip blocks. **Right:** consecutive skipping is disallowed, restricting selections to non-adjacent layers.

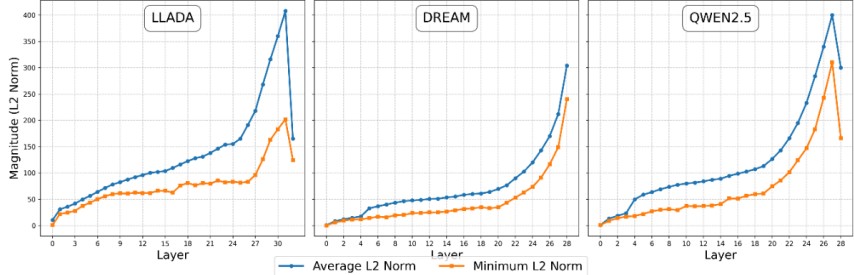

Figure 7: **Hidden-state magnitude across depth.** Layer-wise evolution of the $\ell_2$ norm of token hidden states for LLADA, Dream, and Qwen. Norms remain relatively stable through the first $\sim$60–70% of layers and increase sharply near the top of the network. The *maximum* norm is dominated by rare *sink tokens* (spikes; often $\geq 10^3$), so max values should be interpreted as outliers rather than typical token magnitudes.

lower similarity, reflecting task-specific refinement and decoder-like behavior where representations are actively updated for generation.

In stark contrast, from Fig. 11 reveals that Dream-7B maintains significant recency bias across *all* layers. Consecutive token representations show substantial changes throughout network depth, mirroring the incremental token-by-token refinement characteristic of autoregressive models. This pattern persists despite diffusion training, providing mechanistic evidence that AR initialization creates persistent representational structure. The lack of hierarchical abstraction—with similar update patterns across all depths—explains Dream-7B's brittleness under layer skipping (Table 2), where it behaves more like Qwen2.5 than LLaDA.

## 10.3 LAYER-WISE TOKEN SIMILARITY BY DEPTH

Figures below show token-wise similarity patterns grouped by network depth, revealing the transition from global to local representations:

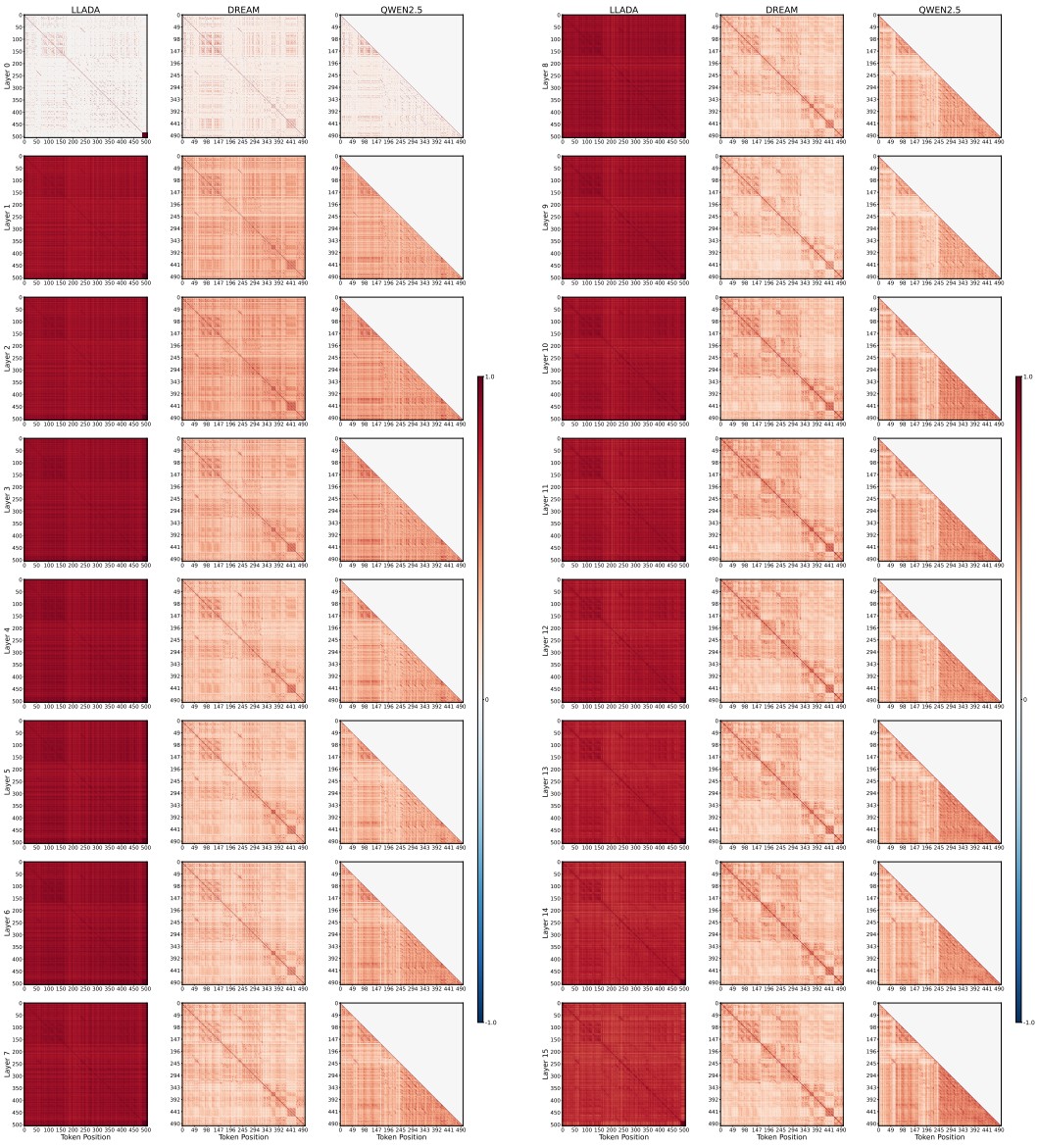

(a) **Token-wise similarity in early layers (0–7).** LLaDA shows uniformly high similarity across all tokens, indicating stable global representations. Dream-7B and Qwen2.5 exhibit lower similarity with visible recency effects, demonstrating incremental AR-style processing even in early layers.

(b) **Token-wise similarity in early-middle layers (8–15).** LLaDA maintains high similarity, while Dream-7B and Qwen2.5 continue showing strong recency bias. The divergence between native dLLM and AR-initialized models becomes more pronounced.

Figure 8: **Token-wise similarity across early and early-middle layers.** (a) Layers 0–7 and (b) layers 8–15.

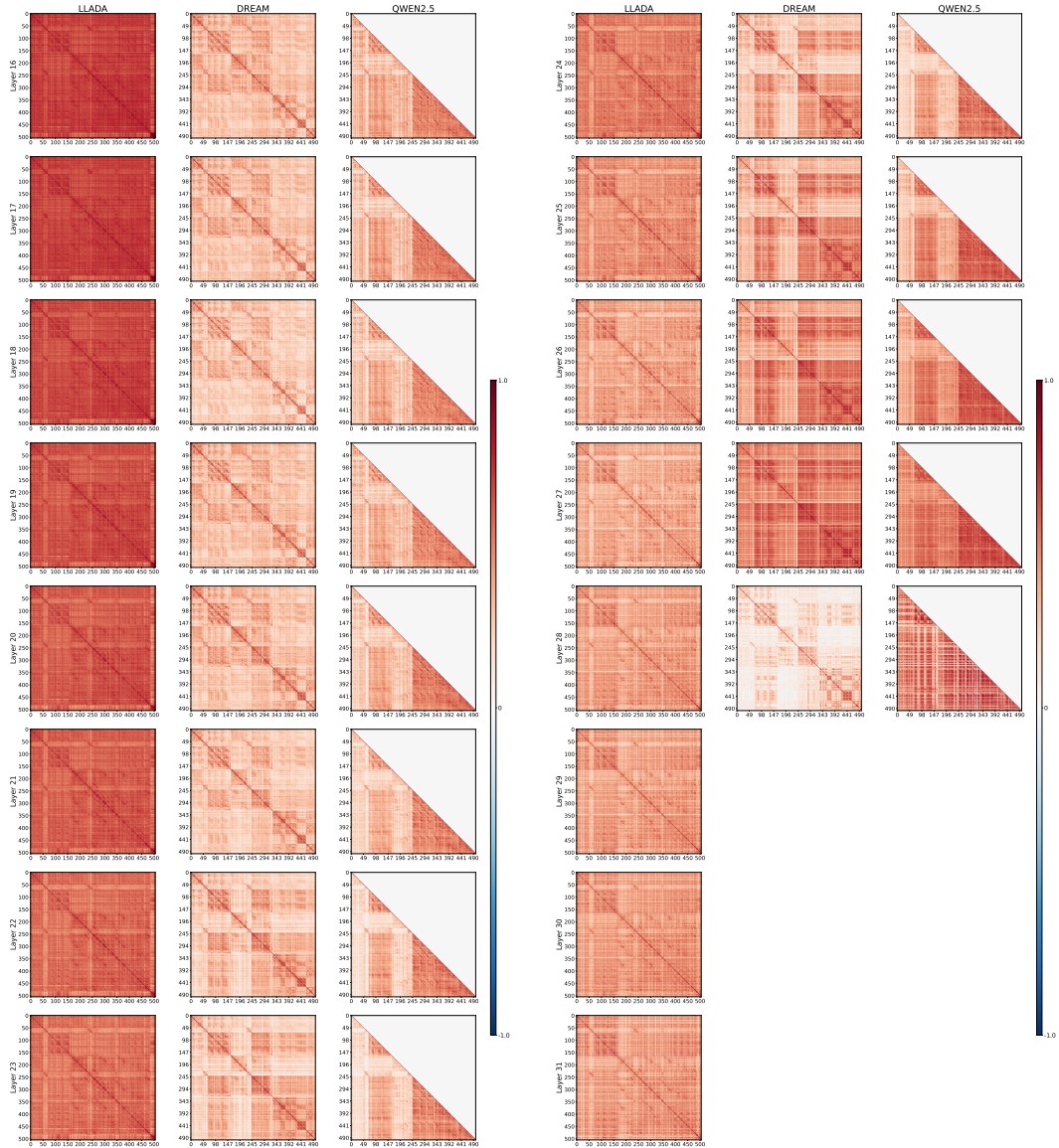

(a) **Late-middle layers (16–23).** LLaDA begins transitioning to lower similarity, indicating the onset of task-specific refinement. Dream-7B and Qwen2.5 maintain consistent recency patterns throughout depth.

(b) **Late layers (24–31).** LLaDA shows increased variability and lower similarity, reflecting active decoder-like refinement for generation. Dream-7B and Qwen2.5 continue incremental updates with strong recency bias, lacking the global transition observed in native dLLMs.

Figure 9: **Token-wise similarity in later layers.** (a) Late-middle (16–23) and (b) late (24–31).

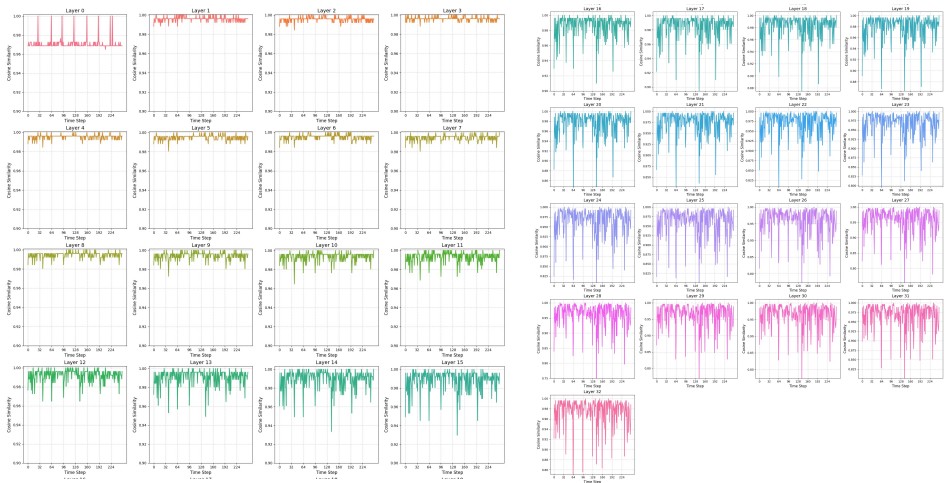

Figure 10: **Token-wise cosine similarity across all layers for LLaDA.** Each subplot shows the cosine similarity between consecutive token representations ($\mathbf{h}_{\ell,i}$ and $\mathbf{h}_{\ell,i+1}$) within a specific layer $\ell$. High similarity indicates smooth representational transitions, while low similarity indicates significant representational changes between tokens. LLaDA exhibits consistently high token-wise similarity across early layers, demonstrating minimal recency bias and global representational abstraction. Later layers show increased variability, indicating task-specific refinement and decoder-like behavior. This pattern validates our hypothesis that native dLLMs develop coarse-to-fine abstraction hierarchies, with early layers establishing stable global context and later layers performing iterative refinement.

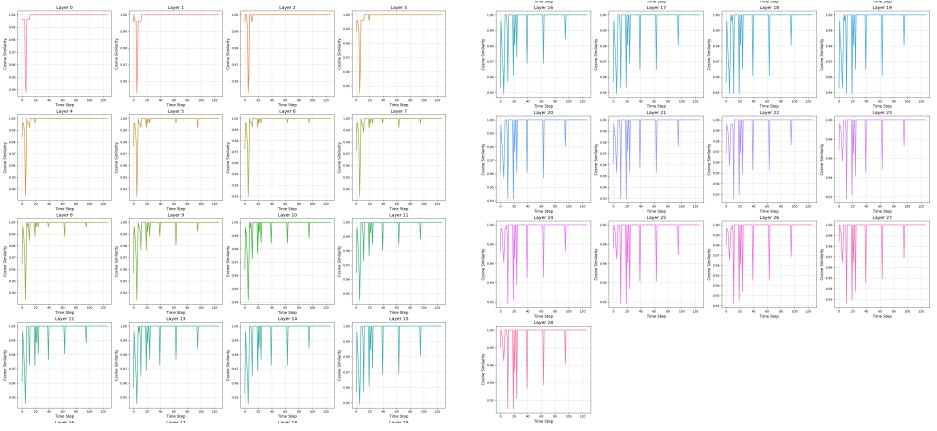

Figure 11: **Token-wise cosine similarity across all layers for Dream-7B.** Each subplot shows the cosine similarity between consecutive token representations within a specific layer. In stark contrast to LLaDA (Figure 10), Dream-7B exhibits significant recency bias across *all* layers, with substantial representational changes for each new token often. The lack of hierarchical abstraction—with similar token-by-token update patterns across all depths—confirms that Dream-7B retains AR-like incremental refinement and retain different representational abstraction compared to native dLLMs. This provides mechanistic evidence for the initialization bias observed in our layer-skip experiments (Table 2), where Dream-7B's brittleness mirrors Qwen2.5 despite diffusion training.

