# OpenReview forum: "Skip To The Good Part: Representation Structure & Inference-Time Layer Skipping in Diffusion vs Autoregressive LLM"
_ICLR.cc/2026/Workshop/Sci4DL — Sci4DL 2026_

### Official Review · Reviewer_haKx · 2026-02-10

**Fit:** 3
**Significance:** 3
**Confidence:** 2

**Summary:**

The paper studies the question: "how do the representations of autoregressive LLMs, diffusion LLMs (dLLMs), and dLLMs initialized from autoregressive LLMs differ"? They show that dLLMs (LLADA) initialized from scratch have coarse-to-fine representations across layers where early layers' representations are highly redundant (high token-wise cosine similarity) while later layers perform refinement and thus have much different representations layer to layer, while autoregressive LLMs (Qwen 2.5) and dLLMs initialized from an autoregressive LLM (Dream 7B) both have layer-to-layer very different representations (so that this "refinement phase" occurs over the whole network architecture). Based on this finding, the authors propose a mechanism to skip the first few layers in native dLLMs to make the model more efficient while minimizing quality loss, measuring 18.75% FLOP reduction in forward pass while retaining 90% of the performance on math/knowledge benchmarks.

**Strengths:**

- The idea is innovative; an analysis of the representations in dLLMs seems to be novel, and the algorithm for adaptively skipping initial layers also seems relatively novel as well as having solid empirical motivation.
- The empirical performance of the algorithm seems good and the empirical analysis of the algorithm seems very thorough; both the algorithm formal statement and all empirical analysis are in the appendix.

**Suggestions:**

- One hypothesis that may be interesting to try is as follows: dLLM is natively trained with a very large "effective depth" (e.g. total depth = # layers in denoiser * # denoising steps), hence having some redundancy in the early layers is relatively forgiveable. Thus it may be the case that deeper LLMs also have some layers emit redundant representations. In this case it could be the "effective depth" near initialization that governs the redundancy of the representation. To verify or reject this hypothesis you could probe some deeper LLMs.
- In terms of language it is not clear to me why "refinement" phase has non-redundant representations, to me it seems the opposite (carefully refining the features should produce similar features at each successive layer). Even though this is just a quirk of English, maybe consider re-naming the phase?

---

### Meta-Review · Area_Chair_a4Te · 2026-03-02

**Recommendation:** Accept

**Metareview:**

Based on the review I recommend accept.

---

### Decision · Program_Chairs · 2026-03-02

Accept